# Atomic-like charge qubit in a carbon nanotube enabling electric and magnetic field nano-sensing

I. Khivrich[1] & S. Ilani [1✉]

Quantum sensing techniques have been successful in pushing the sensitivity limits in numerous fields, and hold promise for scanning probes that study nano-scale devices and materials. However, forming a nano-scale qubit that is simple and robust enough to be placed on a scanning tip, and sensitive enough to detect various physical observables, is still a great challenge. Here, we demonstrate, in a carbon nanotube, an implementation of a charge qubit that achieves these requirements. Our qubit's basis states are formed from the natural electronic wavefunctions in a single quantum dot. Different magnetic moments and charge distributions of these wavefunctions make it sensitive to magnetic and electric fields, while difference in their electrical transport allows a simple transport-based readout mechanism. We demonstrate electric field sensitivity better than that of a single electron transistor, and DC magnetic field sensitivity comparable to that of NV centers. Due to its simplicity, this qubit can be fabricated using conventional techniques. These features make this atomic-like qubit a powerful tool, enabling a variety of imaging experiments.

[1] Department of Condensed Matter Physics, Weizmann Institute of Science, Rehovot 76100, Israel. ✉email: shahal.ilani@weizmann.ac.il

Ultrasensitive nanoscale detectors of electric and magnetic fields take an increasingly central role in advancing the research of devices and materials. When used as scanning probes, these detectors provide a unique insight to electronic and spin systems on the nanoscale. To date, a large variety of scanning probe sensors have been developed, optimized to measure specific physical quantities: Magnetic fields are primarily imaged via scanning SQUIDs[1], Hall probes[2] and NV centers[3], while electric fields are primarily probed with Kelvin probes[4], scanning tunneling potentiometry[5], and scanning single electron transistors (SET)[6]. From all the above techniques, only NV centers utilize a quantum two-level system (qubit) that takes full advantage of the power of quantum manipulations. This gives NV center-based probes unprecedented sensitivity to local magnetic fields, and additionally a modest sensitivity to electric fields[7]. NV scanning probes have also few limitations: optical readout introduces a significant challenge at cryogenic temperatures, and using the sensor at high magnetic fields requires impractical RF frequencies. A different type of a scanning qubit that can sense electric fields ultra-sensitively on the nanoscale and simultaneously probe magnetic fields with modest sensitivity, will have complementary capabilities and is thus highly desirable.

Candidate solid state qubits for nanosensing applications generically divide into two groups—atomic and engineered. Atomic qubits (such as NV centers[8] or P dopants in Si[9,10]) utilize natural atomic wavefunctions as their basis, and hence are small and often have long coherence times. Engineered qubits, on the other hand (e.g., transmon[11] or semiconducting double quantum dot qubits[12,13]) provide finer control over the energy spectrum and the dipole coupling to the physics of interest, however, they are larger (μms to mms), require complex planar circuit designs, and often rely on external detectors for their readout, thus making them less suitable for nanoscale sensing applications. A qubit that can combine the simplicity of atomic qubits with the tunability and control of engineered qubits could therefore lead to a potentially powerful scanning nanosensor.

A conceptually simple qubit that may combine the above advantages can be based on the natural electronic wavefunctions of a single quantum dot. Similar to atomic orbitals, such wavefunctions have distinct spatial structure. This structure, however, occurs on much larger spatial scales and can therefore provide larger and more tunable electric moments. Carbon nanotubes present an excellent setting for realizing this concept; in their recent generations they are electronically pristine, allowing the creation of quantum dots with exceptional level of control over their wavefunctions and energy spectrum[14,15]. So far, engineered double quantum dot qubits have been demonstrated successfully in carbon nanotubes[16–19], but the lithographic complexity of these devices and their frequent reliance on external readout elements such as on-chip resonators, may be prohibitive for using them in scan probes. Moreover, most of these qubits were intentionally designed to be insensitive to external fields and thus are poor sensors. At the same time, single quantum dot devices in carbon nanotubes have been successfully used as ultra-sensitive scanning SETs, allowing to image oxide interfaces[20], Wigner crystals[21], as well as the mapping of ballistic[22] and hydrodynamic[23] electron flows, demonstrating the compatibility of these devices with scanning probe applications.

In this work, we realize a qubit in a carbon nanotube, and demonstrate its application as a highly-sensitive electric and magnetic fields nano-sensor. Our qubit has a simple built-in transport-based readout, a highly local response to electric fields, which we image directly using capacitive techniques, and simplicity that allows placing it at the edge of a scanning probe cantilever. We determine its decay and dephasing times using time-domain and Landau–Zener–Stuckelberg[22] interferometry

experiments and show that its coherence-limited transition leads to significantly improved electric potential sensitivity as compared to the thermally-broadened Coulomb blockade peak of an SET. Furthermore, we demonstrate that the same qubit can simultaneously detect magnetic fields parallel to the nanotube axis. Although the short coherence time did not allow us to implement dynamic decoupling protocols and compete with the AC sensitivity of NV centers, we achieve DC magnetic field sensitivity that is on par with that of NV center[24] and Hall bar-based[25] scanning probes.

## Results

**Forming a qubit using natural wavefunctions in a nanotube.** The basis of our qubit is given by two electronic wavefunctions in a single quantum dot, formed in a suspended carbon nanotube. Within the single-particle picture, a parabolic confinement potential along the nanotube leads to a ladder of harmonic oscillator levels, whose wavefunctions' extent along the nanotube axis increases with increasing level number (Fig. 1a, gray illustrations). Each level is 4-fold degenerate, due to the spin ($\uparrow$, $\downarrow$) and valley ($K$, $K'$) degrees of freedom. In a gapped nanotube, $K$ and $K'$ electrons rotate in opposite directions around the nanotube circumference, leading to opposite orbital magnetic momenta. Applying a magnetic field parallel to the tube axis, $B_{||}$, breaks the spectrum into four independent ladders with slopes given by the orbital and spin magnetic moments, $\frac{\partial E}{\partial B_{||}} = \pm\mu_{spin} \pm \mu_{orb}$, with $\mu_{orb} \gg \mu_{spin}$ (red and blue lines in Fig. 1a correspond to $K$ and $K'$ states). Spin-orbit coupling splits[26] the 4-fold degeneracy at $B_{||} = 0$, and Coulomb repulsion changes the simple non-interacting wavefunctions into Wigner crystals with finer real-space structures[21], yet, since the valley remains a good quantum number, the simple picture in which tuning $B_{||}$ leads to crossing between levels with different magnetic moments and different spatial structures remains valid for the discussion below.

To make an atomic-like qubit that is sensitive to local electric and magnetic fields, we choose a crossing between a high-lying $K$ state and a low-lying $K'$ state with opposite spin directions, which we will denote as $K_n$ and $K'_m$ ($n \gg m$). The charge density of the $K_n$ state is spatially extended (Fig. 1b. left, red) whereas that of the $K'_m$ state is spatially localized (Fig. 1b. left, blue), endowing the qubit transition a localized electric moment. Contrary to a standard charge qubit in double quantum dots, whose charge is localized on the left or right dots, separated by lithographic dimensions, in our case the electrical moment results from the difference in charge distribution of different wavefunctions within a single quantum dot. Due to symmetry of the charge distributions around the center of the dot, the dipole moment of the qubit is approximately zero, potentially reducing its sensitivity to homogenous electric fields and far-field noise, however, this qubit has quadrupole or higher moments that yield strong sensitivity to local fields, which is beneficial for high resolution imaging. The $K_n$ and $K'_m$ states have also opposite orbital momenta (Fig. 1b, center), endowing the qubit transition also a large magnetic moment ($\sim 20\,\mu_B$, $\mu_B$ is the Bohr magneton). We choose opposite spins for the basis states to minimize their overlap, which leads to long decay times.

The schematic charge stability diagram for a hole-doped nanotube single quantum dot is plotted in Fig. 1c as a function of gate voltage, $V_G$, and $B_{||}$. The diagram is obtained by adding the charging energy $U$ to single-particle energies in Fig. 1a, inset. Transport occurs along Coulomb blockade (CB) charging lines, across which a hole is added to the system, zig-zaging between $K$ (red) and $K'$ (blue) character as a function of $B_{||}$. The relevant triple point for our experiment separates the state $|N\rangle$, having $N$ holes, from the two qubit states, $|B\rangle = |N\rangle + K_n$, and

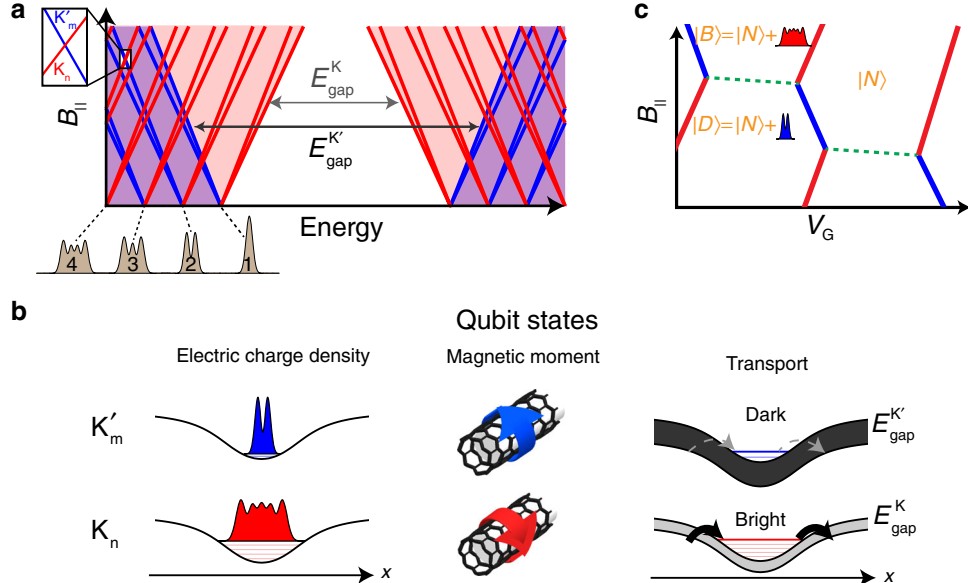

**Fig. 1 A qubit based on the natural wavefunctions in a carbon nanotube. a** Single particle energy spectrum of a single quantum dot in a carbon nanotube, as a function of magnetic field parallel to the tube axis, $B_\parallel$. Each spatial wavefunction (gray illustrations, bottom) has at $B_\parallel = 0$ a 4-fold spin ($\uparrow$, $\downarrow$) and valley ($K$—red, $K'$—blue) degeneracy, which is lifted at finite $B_\parallel$ with slopes given by the spin and orbital magnetic moments $\pm\mu_{spin} \pm\mu_{valley}$. $B_\parallel$ also modifies the bandgap of the two valleys: $E_{gap}^{K}$ decreases and $E_{gap}^{K'}$ increases with $B_\parallel$ (gray arrows). The intersection of two levels at finite $B_\parallel$ (inset) is used as the basis of our qubit. **b** The two intersecting levels (referred to as $K_n$, $K'_m$) differ in three quantities: Since the $K_n$ state is a much higher bound state in the confinement potential than the $K'_m$ state, it is spread more along the nanotube (left). The $K_n$ and $K'_m$ states originate from opposite valleys with opposite magnetic moments due to opposite directions of electron motion around the nanotube circumference (center). The barriers of the dot are formed by the nanotube bandgap, which at finite $B_\parallel$ is different for the two valleys (see panel **a**), making the tunneling from the leads into the $K_n$ state much faster than to the $K'_m$ state (bright/dark, right panel). **c** Charge stability diagram, obtained by adding the charging energy to the energy spectrum in the inset in panel **a**. Coulomb blockade peaks zig-zag between charging of the two valleys (red/blue). The triple point used in the experiment is between the $|N\rangle$ state, having N holes, and the $|D\rangle$ and $|B\rangle$ states, which are obtained by adding either a $K_n$ hole (red) or a $K'_m$ hole (blue).

$|D\rangle = |N\rangle + K'_m$, both having $N+1$ holes. Within the Coulomb valley there should be a boundary line separating the $|B\rangle$ and $|D\rangle$ ground states (dashed green) along which the ground state changes its last occupied wavefunction while maintaining the total charge in the dot fixed. This line will however be invisible in transport, as the system is in Coulomb blockade.

**Experimental realization and transport signature.** We implement the atomic-like qubit in a device that has an single carbon nanotube, assembled at the edge of a cantilever (Fig. 2a). The nanotube is suspended over a distance of 1.2 μm between two Au contacts (S,D) over an array of seven individually controlled gates (Fig. 2a, inset). The height difference between the contacts and the gates (suspension height) is 60 nm. Note that this geometry is identical to our scanning nanotube-based SET cantilever geometry, which we previously used to image 1D[19] and 2D[20,23,27] systems, making the technique developed here directly applicable for scanning probe applications. The device is fabricated using a nano-assembly technique[15], in which nanotube growth and lithography are performed independently, allowing deterministic assembly of a nanotube with desired properties on a complex circuit. In this paper we will use the multiple gates to directly image the charge density of the electronic wavefunctions that form the basis for a qubit. However, we want to emphasize that the qubit that we demonstrate here does not require multiple gates and should work equally well in the simplest single-gated nanotube transistor device which can be formed by standard device fabrication techniques[14]. Our device is cooled in a dry dilution refrigerator, with an electron temperature of $T_{el} \sim 60$ mK as measured by the width of CB peaks, and with a magnetic field parallel to the nanotube axis. The nanotube conductance, G, is measured at zero DC $V_{sd}$ bias using an LC tank circuit connected to the drain contact[28], and with a small AC excitation on source contact (~15 μVrms) at the tank circuit resonant frequency (Supplementary Note 1).

Figure 2b shows G, measured as a function of a common gate voltage, $V_G$, applied together on all gates, and $B_\parallel$. The expected zig-zag behavior of the CB peaks is clearly visible, however, while the $K$ transitions exhibit finite conductance at the CB peak (bright), the $K'$ transitions have no observable conductance (dark), and are marked in the figure by dashed blue lines. Similar dark/bright behavior of the two valleys at finite $B_\parallel$ was observed previously[29,30]. The triple point used for our experiments is shown in the zoom-in measurement (Fig. 2c) with the three relevant ground states, $|N\rangle$, $|B\rangle$ and $|D\rangle$, labeled. To clarify, the 'dark' states described here are not equivalent to the recently reported dark states due to coherent population trapping[31], which occur at finite $V_{sd}$, and do not require $B_\parallel > 0$.

At finite $B_\parallel$, the $p-n$ junction barriers that confine the holes in the two valleys differ significantly (Fig. 1b right). Their height and spatial extent, given by the nanotube bandgap, decreases with $B_\parallel$ for the $K$ states ($dE_{gap}^{K}/dB_\parallel = -2\mu_{orb}$) and increases with $B_\parallel$ for the $K'$ states ($dE_{gap}^{K'}/dB_\parallel = 2\mu_{orb}$), leading to markedly different transport for the two states. The difference in transport visibility between the $|N\rangle \leftrightarrow |B\rangle$ and $|N\rangle \leftrightarrow |D\rangle$ transitions thus gives a built-in transport-based readout mechanism for the qubit state, which does not require an external charge detector.

**Spatial mapping of charge distributions.** In order to form a charge qubit, sensitive to its electric environment, its basis states ($|B\rangle$ and $|D\rangle$) should differ in their charge distribution along the nanotube. We image these charge distributions directly using the

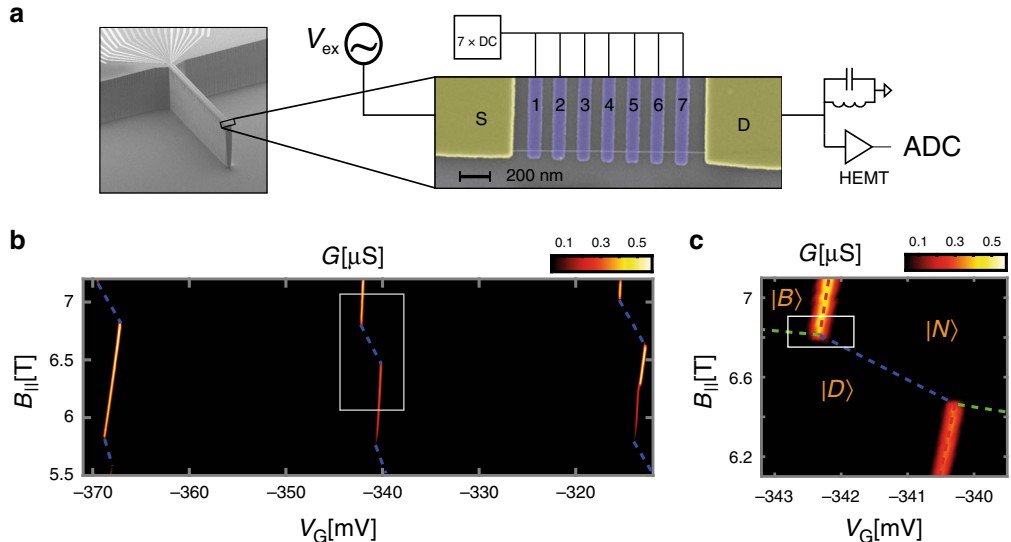

**Fig. 2 Experimental setup and transport signature. a** Scanning electron microscope image of the device: At the edge of an etched cantilever, a nanotube is positioned on two contacts (S,D) and suspended over a piano of seven gates. The conductance through the device is measured with an AC excitation on S ($V_{ex}$ at ~1.5 MHz) and a cryogenic LC tank circuit connected to the drain (D) followed by a cold HEMT amplifier. **b** Measured conductance, $G$, as a function of gate voltage common to all seven gates, $V_G$, and $B_{||}$, exhibiting a zig-zag of bright and dark charging lines (the latter marked by dashed blue). **c** Zoom-in on one triple point, with the schematic lines separating the $|N\rangle$, $|D\rangle$ and $|B\rangle$ states are marked as in Fig. 1c.

array of gates, as follows; First, we tune the voltage common to all gates, $V_G$, to observe the Coulomb peak at the $|N\rangle \leftrightarrow |B\rangle$ transition (illustrated in gray, Fig. 3a bottom). Then, we repeat this scan but with a voltage offset $\Delta V$ added to gate $i$. This will lead to a shift in the Coulomb blockade peak by $\delta V_i$ along the $V_G$ axis, proportional to the local charge density just above this gate (Colored curves in Fig. 3a bottom, Supplementary Note 2). By measuring the individual shifts with respect to all gates, $\delta V_i$, $i = 1..7$, we thus image the charge density added on the transition at seven spatial points, which is essentially the discrete version of the scanning imaging of Wigner crystals that we performed previously[21]. Although the $|N\rangle \leftrightarrow |D\rangle$ transition is dark in transport, we still know its position accurately by connecting the corners of the bright transitions (dashed blue, Fig. 2c). Thus, using the same method we can also image the spatial charge density within the $|D\rangle$ state.

Figure 3b zooms in on the triple point around $|N\rangle$, $|B\rangle$ and $|D\rangle$ (white square in Fig. 2c). Upon addition of $\Delta V = 0.5\,mV$ to gate 4 the bright and dark transition shifts by independent amounts, $\delta V_4^B$ and $\delta V_4^D$ (gray arrows). Similar measurements with all gates yields the shifts $\delta V_i^B$ and $\delta V_i^D$, which when plotted as a function of gate position (Fig. 3c and d) trace the spatial distribution of the charge added at the $|N\rangle \leftrightarrow |B\rangle$ and $|N\rangle \leftrightarrow |D\rangle$ transitions, $\rho_{NB}(x)$ and $\rho_{ND}(x)$, where $x$ is the spatial coordinate along the nanotube. Visibly, while $\rho_{NB}(x)$ is homogenously spread over all gates, $\rho_{ND}(x)$ is localized at the dot's center.

**Time-domain measurements.** To study the dynamics of a $|D\rangle$, $|B\rangle$ qubit we turn to time domain experiments that use gate voltage on the central three gates, $V_G$, as a fast control axis, with the following sequence: First, the dot is initialized in the $|B\rangle$ state, on the $|N\rangle \leftrightarrow |B\rangle$ Coulomb peak ($V_G = V_{CB}$, black star, Fig. 4a). Then, a fast ramp to $V_G = V_{probe}$ is applied, after which the system is left to evolve for time $\tau_{probe}$. Finally, the voltage is swept back to the initial CB point for readout, dwelling for time $\tau_{read,init}$. If after the probing stage the system ended up in the ground state $|B\rangle$, the dot will freely conduct in the readout stage. However, if the system switched to the excited state $|D\rangle$, it will remain in the dark state during readout, blocking the conductance. The characteristic blocking time is given by the

fastest of two possible decay routes, $|D\rangle \rightarrow |N\rangle$ or $|D\rangle \rightarrow |B\rangle$, both of which initialize the system to its ground state. The above sequence is repeated periodically, and we measure the conductance averaged over this sequence, which contains two terms: $\langle G \rangle = \left( G\left( V_{probe} \right) \tau_{probe} + G(V_{CB}) \langle P_B \rangle \tau_{read,init} \right) / \left( \tau_{probe} + \tau_{read,init} \right)$. The first term reflects the conductance measured during the probing stage, and is non-zero only for $V_{probe}$ near the Coulomb peak, where the dot has a finite conductance. The second term reflects the conductance measured in the readout stage, and is directly proportional to the mean bright state probability, $\langle P_B \rangle$ during this stage.

Figure 4b shows $\langle G \rangle$ measured as a function of $V_{probe}$ within the above sequence (blue), using $\tau_{probe} = 0.8\,\mu s$ and $\tau_{read,init} = 5\,\mu s$, as well as the measured quasi-DC conductance, $G$, (red). The CB peak in $G$ appears also in $\langle G \rangle$, as expected from the first term the equation above. Interestingly, however, inside the Coulomb valley $\langle G \rangle$ shows a sharp dip at $V_{probe} = V_{BD} \approx -343$ mV, not present in $G$. This dip is much narrower (~40 $\mu V$) than the thermally-limited CB peak (~200 $\mu V$). From $\langle P_B \rangle$ extracted from $\langle G \rangle$ and $G$ using the equation above (Fig. 4c) we see that for most values of $V_{probe}$ the state remains bright ($P_B = 1$), but at the dip the dark state becomes significantly occupied ($\langle P_B \rangle \approx 0.7$). Repeating the above measurement at various values of $B_{||}$ (Fig. 4d) shows that this dip traces a straight line terminating at the $|N\rangle$, $|B\rangle$, $|D\rangle$ 'triple point', as expected from the $|B\rangle \leftrightarrow |D\rangle$ degeneracy line (dashed green in Fig. 1c). Its finite slope suggests that this transition is sensitive to both local magnetic and electric fields, where the latter attests to the different charge distribution within the $|B\rangle$ and $|D\rangle$ states.

Similarly to Fig. 3, we can directly image the charge density distribution change at the $|B\rangle \leftrightarrow |D\rangle$ transition, by measuring the response of the transition line position, $V_{BD}$, to small gate perturbations. The measured density distribution, $\rho_{BD}(x)$ (Fig. 4e, left) compares well to difference between the bright (Fig. 3c) and dark (Fig. 3d) state densities, $\rho_{BD}(x) \approx \rho_{NB}(x) - \rho_{ND}(x)$ (Fig. 4e, right), further establishing the narrow transition line as the boundary between the $|B\rangle$ and $|D\rangle$ ground states. From the measured $\rho_{BD}(x)$ we see that the qubit charge redistribution is

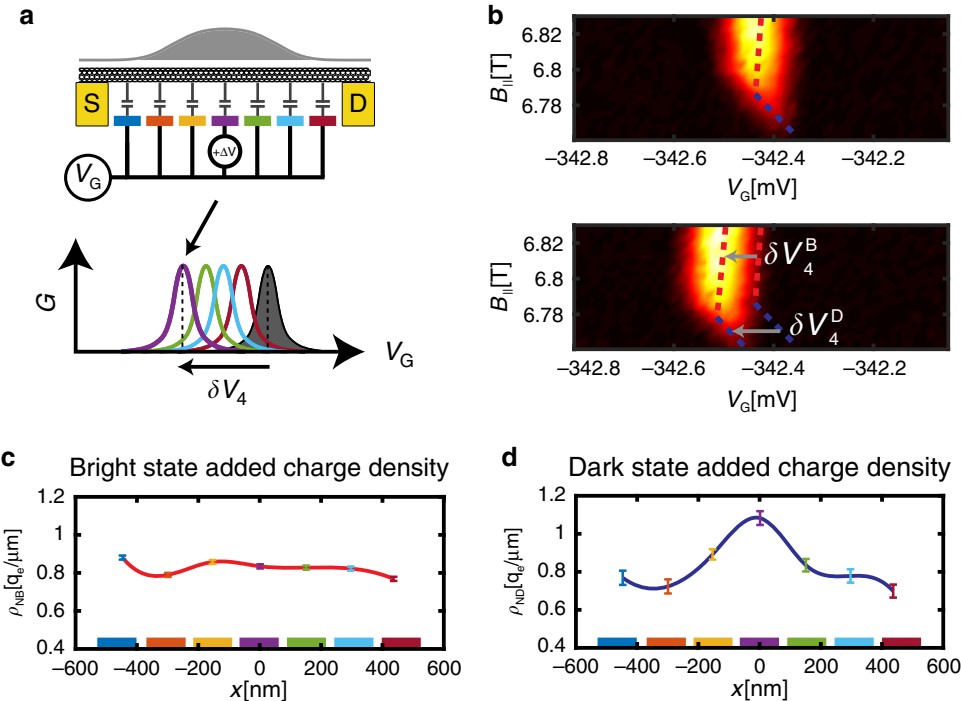

**Fig. 3 Imaging the bright and dark states' spatial charge distributions. a** The charge distribution of a state is imaged by measuring its independent capacitances to each of the individual gates (colored). This is achived by monitoring how the state's Coulomb blockade charging peak (gray, bottom) shifts in common gate voltage, $V_G$, when a small voltage perturbation, $\Delta V$, is added to each of the gates (colored peaks). The individual shifts are directly proportional to the local charge densities above the corresponding gates. **b** Top: Zoom-in on the triple point in the white box in Fig. 2c. Bottom: Same triple point when a perturbation $\Delta V = 0.5$ mV was added to gate 4. Bright and dark states charging lines (dashed red and blue) have independent shifts along $V_G$, labeled $\delta V_4^B$ and $\delta V_4^D$. **c** Measured spatial distribution of the bright state added charge, $\rho_{NB}(x)$, where $x$ is the spatial coordinate along the nanotube. Colored points (with 1$\sigma$ confidence errorbars) correspond to the voltage shifts measured with the corresponding gates (colored gates are plotted along the $x$ axis at their actual spatial cordinates). The red spline connecting the measured points is a guide to the eye. Notably, the bright state charge is spread rather homogenously along the suspended nanotube. **d** Similarly measured spatial distribution of the dark state added charge, $\rho_{ND}(x)$, observed to be concentrated at the center of the confinement well.

also narrow in space. The observed width (~200 nm, Fig. 4a) is limited by the resolution of the imaging method due to the size and distance to the gates. A more quantitative analysis that deconvolves the known shape of the potential distribution produced by the gates concludes that the actual width is ~100 nm (Supplementary Note 4). This width sets the spatial resolution of the qubit sensor.

To measure the transition rate ($T_1$ time) and its dependence on the detuning from the degeneracy point, $V_{BD}$ (Fig. 5a), we repeat the measurements above but with different dwell times, $\tau_{probe}$. In Fig. 5b we plot $\langle P_B \rangle$ as a function of the voltage offset, $\Delta V_G = V_{probe} - V_{BD}$, for various $\tau_{probe}$ values, and in Fig. 5c we plot it as a function of $\tau_{probe}$ for different values of $\Delta V_G$. Away from the dip, the decay time is too long to reliably be extracted from this figure, whereas at the dip it becomes significantly shorter, $T_1 \sim 1$ μs, indicating a fast transition from $|B\rangle$ to $|D\rangle$.

**Coherent behavior and estimating system parameters**. To observe quantum coherence of the $|D\rangle$, $|B\rangle$ qubit and estimate its $T_2^*$, we use Landau-Zener-Stuckelberg (LZS) interference[22,32]. In this case, instead of waiting at $V_{probe}$ for $\tau_{probe}$, the detuning is steered as $V_G(t) = V_{BD} + \Delta V_G + A_{LZS} \sin(2\pi f_{LZS} t)$ (Fig. 5d) and we probe $\langle P_B \rangle$ after time $\tau_{probe}$ by moving $V_G$ to the CB peak and measuring $\langle G \rangle$ as before. The $\langle G \rangle$ measured as a function of $\Delta V_G$ and $A_{LZS}$ at a frequency of $f_{LZS} = 0.7$ GHz (Fig. 5e) shows the characteristic LZS interference pattern. The peak width, $\delta\omega \sim 2\pi \times 180$ MHz, indicates that the qubit maintains coherence over several oscillations, having a $T_2^*$ time of ~0.9 ns.

The observations can be quantitatively explained by a simple model, describing the evolution of the system in the $|D\rangle$, $|B\rangle$ manifold; The unitary evolution is described by the Hamiltonian $H = \epsilon(t)\sigma_z + \Delta\sigma_x$, where the $|D\rangle$ and $|B\rangle$ are the eigenvectors of $\sigma_z$. The dominant decoherence mechanism with rate $\gamma_2$ results from coupling to charge noise acting along the energy detuning axis, $\epsilon(t)$, coupling only to $\sigma_z$. In the far-detuned regime ($\epsilon \gg \Delta, \gamma_2$), the noise changes the phase difference between the basis states ($T_2$ processes), however, close to zero detuning it translates to incoherent transition rate between the basis states ($T_1$ process) (Supplementary Note 5). Consequently, the transition rate between the states depends sharply on the detuning (Fig. 5a, bottom). Using this model we quantitatively fit the results in Fig. 5b, c (solid lines) and obtain the qubit's splitting, $\Delta = 2\pi \times 2$ MHz, and its decoherence rates $\gamma_1 = 2\pi \times 1.5$ kHz, $\gamma_2 = 2\pi \times 185$ MHz (details in Supplementary Note 6), where $\gamma_1$ results from the noise coupled through $\sigma_x$, $\sigma_y$. With the same parameters we also reproduce (Fig. 5f), quantitatively well the LZS measurements in Fig. 5e (Simulation details in Supplementary Note 13).

**Estimating performance of the device as a local sensor**. The strong dependence of the qubit transition on electric potential and $B_{||}$ implies that it can serve as an excellent nano-scale probe of these quantities. Since SET is the most sensitive scanning electrometer to date, we benchmark the qubit sensitivity against measurements in SET modality. DC electric potential sensitivity is measured by slowly ramping the central gate voltage ($V_4$) up and

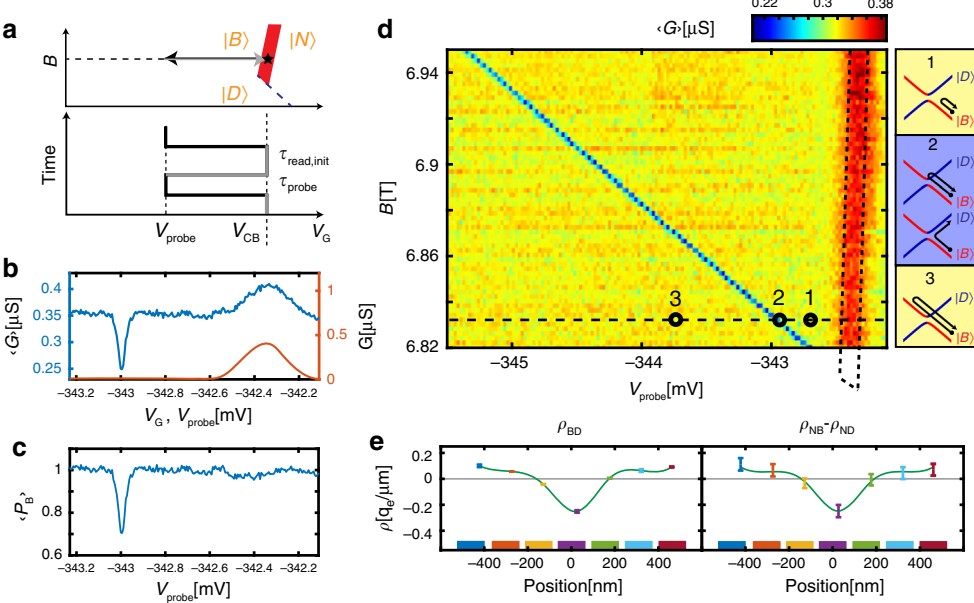

**Fig. 4 Time domain measurements of the qubit transition. a** Measurement sequence, plotted in gate voltage–$B_{||}$ plane (top) and in gate-voltage time diagram (bottom). The state is initialized on the $|N\rangle \leftrightarrow |B\rangle$ Coulomb blockade (CB) line (red, star marks initialization point). Voltage on the center three gates is rapidly ramped to a value $V_{probe}$, dwelling a time $\tau_{probe}$, and then ramped back to the original point on the CB line, dwelling a time $\tau_{read,init}$. The sequence is repeated periodically. **b** Conductance averaged over the above sequence, $\langle G \rangle$, measured as a function of $V_{probe}$ (blue) and the quasi-DC conductance, $G$(red), measured at the same point in $V_G$. **c** Bright state return probability, $\langle P_B \rangle$ as a function of $V_{probe}$, determined from $\langle G \rangle$ and $G$ in panel **b** (see text). **d** Similar measurement of $\langle G \rangle$ but now as a function of $V_{probe}$ and $B_{||}$. A red peak is visible at the location of the Coulomb blockade peak and a much narrower blue dip appears along a line that corresponds to the $|D\rangle \leftrightarrow |B\rangle$ crossing. Both a fast traverse to a point before this line (marked 1) or after this line (marked 3) do not alter the state of the system from its initialized $|B\rangle$ state, however, a fast traverse to a point on the line (marked 2) results in occupation of the $|D\rangle$ state with a significant probability. The corresponding traverses in energy are shown in the side panels. **e** Left: The change in spatial charge distribution at the $|D\rangle \leftrightarrow |B\rangle$ transition, $\rho_{BD}(x)$, imaged using gate resolved capacitance shift imaging of this line (as outlined in Fig. 3a). Right: The difference between the the spatial charge distributions measured at the $|N\rangle \leftrightarrow |B\rangle$ and $|N\rangle \leftrightarrow |D\rangle$ transitions, $\rho_{NB}(x) - \rho_{ND}(x)$, taken from Fig. 3c and d, strongly resembling the directly imaged $\rho_{BD}(x)$ in the left panel, further confirming that the narrow line corresponds to the $|D\rangle \leftrightarrow |B\rangle$ transition.

down, and monitoring conductance changes in the two modalities: On the qubit $|B\rangle \leftrightarrow |D\rangle$ transition line using fast gating, and on the SET $|N\rangle \leftrightarrow |B\rangle$ CB line within the same triple point. Parameters are optimized separately for each modality, and the results are converted to a common potential scale (Fig. 6a). Visibly, the qubit provides a significantly improved sensitivity, primarily due to its sharper transition line as compared to the temperature-limited Coulomb blockade peak. The sensitivity to detuning that we obtain in the qubit measurements is ~60 neV Hz$^{-0.5}$ which translates to a potential sensitivity of ~600 nV Hz$^{-0.5}$ (Supplementary Note 8), significantly improving over the performance of the device as an SET, and surpassing the sensitivity of our best SETs to date[27]. DC $B_{||}$ sensitivity is measured in a similar fashion (Fig. 6b). Here, the advantage of qubit detection as compared to an SET becomes even more evident, reaching a sensitivity of ~39 μT Hz$^{-0.5}$ (Supplementary Note 9), comparable to the DC magnetic field sensitivity of NV centers[24]. What limits the sensitivity in the current experiment is the large contact resistance of our device ($R_c \sim 2$ MΩ) and magnetic field fluctuations inherent to a magnet power supply. Theoretical estimates predict that the performance can be improved by more than an order of magnitude by improving the contact resistance and using a persistent mode magnet (Supplementary Note 7). An additional important difference between the two modalities is their back-action on the measured system. While the SET will fluctuate between two states with different charge values on the dot, in the qubit modality, only a small redistribution of the charge along the axis of the nanotube will occur (Fig. 3d). This translates into a reduced back-action of the measurement. In principle by modifying the electrical moments of the basis states

of the qubit one can continuously tradeoff sensitivity for reduced backaction. This feature is extremely important in measurements of fragile quantum states of matter[21].

Our sensor requires finite $B_{||}$ for its operation; however, it can operate in a wide range of magnetic fields (demonstrated at 3–8T, see Supplementary Note 12). This provides complementary capability to that of scanning SQUIDs and NV centers, which generally work only at lower fields, although achieving better magnetic field sensitivities. Importantly, the field direction required to tune the sensor is in-plane for the scanned system, and will couple to electron in a probed 2D sample through a rather small Zeeman energy shift ($g\mu_B B \sim 350$ μeV at 3T), which would be negligible for many of the interesting phenomena in 2D. Independently, an out of plane magnetic field component can be applied to tune the properties of the scanned 2D system. The measurement requires dilution temperatures, however, the sample under study can be thermally decoupled[23] from the probe, and its temperature can in principle be tuned over a large temperature range while keeping the qubit cold, by using local heating only of the electron system[33].

The spatial resolution demonstrated here (~100 nm) was limited by the rather long (1.2 μm) device used in this study to enable the gate imaging in Fig. 3. In principle, it should be straightforward to implement the same qubit in a much shorter and simpler, single-gated suspended device and the resolution will scale in proportion, to the tens of nm range. The geometry of the current device is equivalent to the standard scanning SET cantilevers, and is thus fully compatible with scanning.

Recent work[34] demonstrated spin qubits in carbon nanotube double quantum dots with significantly improved coherence

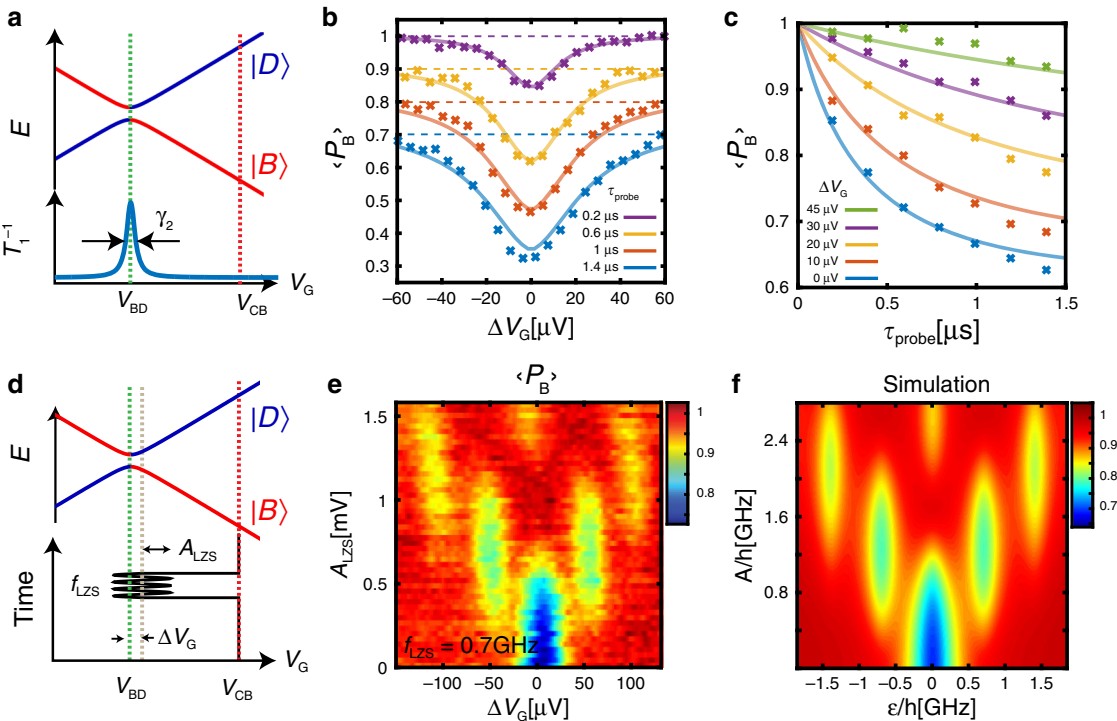

**Fig. 5 Measuring the decay and coherence times of the atomic-like qubit. a** Incoherent transition between the two states of the qubit, $|B\rangle$ and $|D\rangle$, occurs near their degeneracy point in gate voltage, $V_{BD}$. **b** Measured $\langle P_B \rangle$ as a function of $\Delta V_G = V_G - V_{BD}$, for different $\tau_{probe}$. Curves are offseted for clarity, and the dashed colored lines corresponds to $\langle P_B \rangle = 1$ for the corresponding traces. **c** Measured $\langle P_B \rangle$ as a function of $\tau_{probe}$ for few values of $\Delta V_G$. Solid lines in both panels b and c are fits to a model in which the decoherence results from charge noise, described in the main text. All the lines are calculated with a single set of parameters. **d** Landau–Zener–Stuckelberg (LZS) interferometry of the qubit in which a modulation $A_{LZS} \sin(2\pi f_{LZS} t)$ is added to the time sequence in panel a during the probing stage. **e** $\langle P_B \rangle$ measured as a function of $\Delta V_G$ and $A_{LZS}$, with $f_{LZS} = 0.7$ GHz, exhibiting the LZS interference pattern. **f** Simulation reproducing the measurements in panel e (see text), using the same parameter used to reproduce the data in panels **b** and **c**.

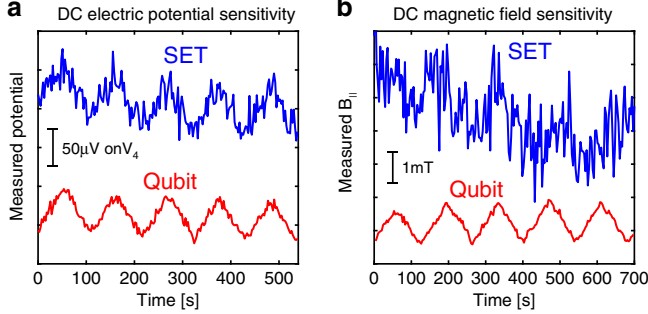

**Fig. 6 Comparing SET and qubit performance in sensing DC electric potential and DC magnetic fields. a** DC electric potential measurement: we slowly ramp the potential of a single gate ($V_4$) up and down (sawtooth) with a period of ~100 s, and probe it using the device in SET modality (blue) or by detecting qubit state transitions using the time domain sequence described in Fig. 4a (red). The potential scale is shown by the vertical bar. **b** Similarly, but now for DC magnetic field measurement. In both quantities, we observe a significant improvement in sensitivity when the device operates in qubit modality rather than SET modality.

times as compared to the results above. However, the sensitivity of these qubits to external magnetic fields is rather limited as compared to that of the atomic-like qubit demonstrated here. The two primary reasons are the exchange coupling to the feromagnetic leads, which results in weak coupling to an external field, and the fact that the magnetic moment difference between the two qubit basis states is that of a single spin, ~30 times smaller

than the magnetic moment difference in the atomic-like qubit, which follows from the orbital moments.

In summary, we have demonstrated a charge qubit in carbon nanotubes that combines the advantages of atomic and engineered qubits. This qubit is conceptually simple, requires only conventional fabrication, has a small form factor allowing placing it on a scanning probe tip, has a simple built-in readout mechanism, and enables sensitive measurements of electric and magnetic fields. These features make it an enabling tool for a variety of experiments. For example, since an atomic-like qubit can be much smaller than lithographic dimensions, which constrain double-dot qubits, it can couple to high vibrational modes of suspended carbon nanotubes, which are at their quantum ground state at dilution temperatures, thus enabling quantum nano-mechanical experiments in this system. As a scanning detector that measures simultaneously electric and magnetic fields, it will be instrumental in exploring phenomena that have both charge and magnetic (/electric current) signatures. Few examples include imaging current whirlpools in hydrodynamic electron flow[35], as well as imaging of quantum flows, including various electron optics phenomena, electron interference, electron localization, magnetic focusing of electrons and of composite fermions. The increased sensitivity and reduced back-action of the qubit will further allow to image the quasiparticles of fragile states of matters, including the observation the electrical charges of topological quasiparticles. More broadly, the addition of quantum sensing and time domain capabilities into scanning electrical field measurements opens the door for sensitivity improvements and for imaging the dynamics in quantum systems, that were so far beyond reach.

## Data availability
The data that support the plots and other analysis in this work are available from the corresponding author upon request.

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

## Acknowledgements
We thank A. Finkler, B. Kalisky, F. Kuemmeth, and E. Zeldov for helpful suggestions. We further acknowledge support from the Minerva grant no. 712290, the Helmsley Charitable Trust grant, and the ERC-Cog (See-1D-Qmatter, No. 647413).

## Author contributions
I.K. and S.I. conceived the experiment. I.K. built the experimental system, performed the experiments, analyzed the data, and preformed the simulations. I.K. and S.I. wrote the manuscript.

## Competing interests
The authors declare no competing interests.
