## [Peer Review File · Nature Communications]

Reviewers' comments:

Reviewer #1 (Remarks to the Author):

Dear Authors,

Please see below for my comments:

What are the major claims of the paper?

Khivrich and Ilani experimentally demonstrate a novel qubit implementation in a carbon nanotube (CNT); the two-level qubit state is chosen at the location of non-zero interactions between "bright" and "dark" valley states, which are brought into degeneracy by applying large magnetic fields (3 to 6 Tesla) along the axis of the CNT.

The negligible electric dipole moment between these two states allows for insensitivity to far-field electric fields, while the non-zero quadrupole moment allows for high-sensitivity sensing of near-field electric fields down to ~ 600 nV/rHz. In the same mode of operation, the large magnetic moment derived from the different valley states in largely disparate occupancies allows for high sensitivity sensing of magnetic fields down to ~ 39 μ T/rHz, which is within an order of magnitude of demonstrations using single NVs in diamond at cryogenic temperatures.

Are they novel and will they be of interest to others in the community and the wider field? If the conclusions are not original, it would be helpful if you could provide relevant references.

The authors are among the world-leading experts at lithographically patterning gates onto low-disordered CNTs, and using such systems to explore a wide range of exotic, correlated phenomena at cryogenic temperatures. While the demonstration of such a qubit state can be considered an interesting operational regime in their experimental system, the broad applicability of such a system remains questionable. Specifically, the strict requirement ($k_B T < \text{orbital/charging/valley energies}$) for their "sensor" to operate at low temperatures means there must be a compelling reason to study any exotic behavior of a sample at cryogenic temperatures. Typical phenomena of interest at low temperatures include magnetic ordering or superconducting properties. However, the requirement for high magnetic fields (> 3 T) would exclude many such samples of interest; for example, superconducting materials eventually quench at high magnetic fields.

While the authors point out imaging studies of "hydrodynamic electron flow, ... topological magnetic structures, and ... multiferroics", it's not clear that a majority of such phenomena would be present under the conditions demanded by the demonstrated qubit state in the CNT.

Is the work convincing, and if not, what further evidence would be required to strengthen the conclusions?

The scientific evidence is quite compelling. However, the authors may consider showing the power spectral density of the measurements in far-detuning and non-detuning modes of SET vs Qubit operation. Seeing the spectral density across the measurable frequency range of the sensor would give a sense of the wider applicability of such a sensor.

On a more technical note, if the lever arm for each gate falls in the range of < 0.2 , then does that suggest a non-zero cross-capacitance from the 7 gates to separate parts of the CNT? If so, how is that taken into account when mapping out the spatial profile of the Bright and Dark wavefunctions?

Also, if the qubit state is as robust as the authors claim, it's unclear to me why the lifetimes ($T_2^* \sim 0.9$ ns and $T_1 \sim \mu$ s) are so short. Have the authors quantitatively studied the effect of magnetic instability and high contact resistances on such values? I'm aware that there's a supplemental section on the theoretical limit on the potential sensitivity, but there's no mention of what would be the expected T_2^* and T_1 of such an ideal scenario.

On a more subjective note, do you feel that the paper will influence thinking in the field? Please feel free to raise any further questions and concerns about the paper.

I believe the technical feat presented in this work is very impressive, and it's quite possible that

such a system will be applicable to studies in the future that are not yet clear to the community now. The novelty and technical feat already justifies publication. While the community would benefit from seeing this technological progression of gate-defined quantum dots in CNTs, it's not entirely clear to me that such a demonstration would be of immediate interest to the general readership of Nature Communication.

We would also be grateful if you could comment on the appropriateness and validity of any statistical analysis, as well the ability of a researcher to reproduce the work, given the level of detail provided.

Perhaps this is discussed in a previous paper by the authors, but it'd be helpful to include some notion of the fabrication yield for producing such CNT-based sensors within a single chip. Or, if there is no way to quantify this yield, how does this challenging fabrication process affect its prospects for wider adoption?

Some typos:

Page 6: "thus, using the same method we can [also] image the spatial..."

Page 6: "this will lead to a shift [in] the Coulomb blockade..."

Page 8: Similarly to Fig 2, we can [directly] image the charge..."

Page 10: "Our sensor requires finite B// for its operation[;] however, it can..."

Respectfully,
A Reviewer

Reviewer #2 (Remarks to the Author):

In this manuscript, the authors propose a scanning probe technique that involves a new type of qubit in a carbon nanotube. This qubit relies on the very special electronic configuration of carbon nanotubes where the valley degeneracy can be split by a parallel magnetic field. This system is thus sensitive to both electric and magnetic fields. The difference of DC conductance in the two qubit states allows a simple readout scheme, relying only on conductance measurements. In this paper, the qubit is characterized and its sensitivity as a sensor is estimated. It proves to be more sensitive to electric field than SET scanning probes and comparable with NV centers for the magnetic field.

On the whole, the article is clearly written and succeeds to explain the principle of the detection. The data presented are clean and look sound. Additional details on various aspects of the experiment are present in the supplementary information. The conclusion of the article, namely that this new kind of qubit is a valuable tool for quantum sensing, is rather convincing and will probably be interesting for the community.

Note that there is no data demonstrating the use of this qubit for sensing of a "real" physical system, such that the point of the article is more applied than fundamental.

I thus recommend publication in Nature Communications, providing that the authors address the minor following points:

- Very few details are given about the fabrication of the sample. The authors refer to a previous work, but it would be beneficial to give some important information in the paper, at least in the supplementary information. For example, what is the material contacting the carbon nanotube? This would be interesting regarding the contact resistance.
- The setup relies on the fact that the valley is a good quantum number, which is still the case in presence of spin-orbit coupling (page 4). However, it is quite common to find mixing between the K and K' valley, even in very clean nanotubes due to boundary conditions (Marganska, PRB 92 (2015)). How difficult is it to find a "good" carbon nanotube? Does this compromise the relative simplicity of the setup?

Here are a few typos I found:

- Caption of fig. 1.b : a space is missing between "into" and "the"
- Page 5 "The nanotube is suspended a distance of 1.2 μm ": I think a word is missing
- Caption of fig. 5: DC electric potential measurement

Reviewer #3 (Remarks to the Author):

The paper "Electric and magnetic field nano-sensing using a new, atomic-like qubit in a carbon nanotube" reports the implementation of a new quantum sensing scheme using transport measurements in an ultra-clean carbon nanotube device. Quantum sensing is a very active field of quantum technology in which the idea is to use the control on individual quantum system as an ultimate probe -ideally quantum limited- for various applications. In particular, scanning probes using such systems are very interesting for probing a variety of condensed matter states. The manuscript describes a conceptually new scheme which uses a rather simple system - single dot in a carbon nanotube with an easy electrical interface. This new "atomic-like qubit" is made possible by the nano-assembly technique pioneered by the group of S. Ilani for making carbon nanotube quantum devices. Carbon nanotubes are very appealing in that context since they are a very good match between mesoscopic devices and very well controlled spectrum. The results demonstrated here provide a proof of principle that this new quantum sensor has a high combined sensitivity to local electric and magnetic fields. It could therefore be very interesting to incorporate it in an actual scanning probe setup.

The results presented here are definitely of high quality and the manuscript well written. I recommend publication in Nature Communications. I would like the authors to consider the following comments/questions:

1. I understand that the device presented in the paper is a proof of principle and that probably many technical developments will have to be sorted out to incorporate it in a scanning setup. However, it would be nice to have more details about how the authors technically envision this, in particular regarding how to bring the K_n/K'_m to degeneracy. Have the authors also thought about the back-action of this sensor on some benchmark physical systems ?
2. What is the physical origin of the avoided crossing Δ ? It does not seem to be mentioned anywhere in the paper.
3. In figure S7, there are clearly additional lines which show that the system departs from the simple picture shown in figure 1b. This could mean (for example if it comes from residual disorder) that the corresponding transition will have an electrical dipole and not only higher order multipoles. It might therefore be sensitive to the non-local part of the electric field. Is it going to be a limiting factor for local electric field sensing ?
4. The high ohmic resistance of the devices made using the nano-assembly technique seems to be a recurrent problem and is a limitation for the sensing capabilities of the atomic-like qubit. How could this be optimized ?
5. Regarding the B-field sensitivity, how would single spin 1/2 detection compare to the actual figures of merit of the atomic-like qubit. Ultra-narrow linewidths of single spin in carbon nanotubes limited by nuclear spins have been recently demonstrated in double dot carbon nanotubes and might be useful as well (see T. Cubaynes et al. NPJQI 5, 47 (2019)).

Reviewer #1:

What are the major claims of the paper?

Khivrich and Ilani experimentally demonstrate a novel qubit implementation in a carbon nanotube (CNT); the two-level qubit state is chosen at the location of non-zero interactions between “bright” and “dark” valley states, which are brought into degeneracy by applying large magnetic fields (3 to 6 Tesla) along the axis of the CNT.

The negligible electric dipole moment between these two states allows for insensitivity to far-field electric fields, while the non-zero quadrupole moment allows for high-sensitivity sensing of near-field electric fields down to $\sim 600\text{nV/rHz}$. In the same mode of operation, the large magnetic moment derived from the different valley states in largely disparate occupancies allows for high sensitivity sensing of magnetic fields down to $\sim 39\mu\text{T/rHz}$, which is within an order of magnitude of demonstrations using single NVs in diamond at cryogenic temperatures.

Are they novel and will they be of interest to others in the community and the wider field? If the conclusions are not original, it would be helpful if you could provide relevant references.

The authors are among the world-leading experts at lithographically patterning gates onto low-disordered CNTs, and using such systems to explore a wide range of exotic, correlated phenomena at cryogenic temperatures. While the demonstration of such a qubit state can be considered an interesting operational regime in their experimental system, the broad applicability of such a system remains questionable. Specifically, the strict requirement ($k_B T < \text{orbital/charging/valley energies}$) for their “sensor” to operate at low temperatures means there must be a compelling reason to study any exotic behavior of a sample at cryogenic temperatures. Typical phenomena of interest at low temperatures include magnetic ordering or superconducting properties. However, the requirement for high magnetic fields ($>3\text{T}$) would exclude many such samples of interest; for example, superconducting materials eventually quench at high magnetic fields.

While the authors point out imaging studies of “hydrodynamic electron flow, ... topological magnetic structures, and ... multiferroics”, it’s not clear that a majority of such phenomena would be present under the conditions demanded by the demonstrated qubit state in the CNT.

We thank the referee for this comment/question. Indeed, in the manuscript we commented only briefly on the type of systems that can be studied by this scanning qubit. As with any other sensor, our qubit is not going to be applicable for studying any generic system, however, we strongly believe that it can be applied to a wide spectrum of interesting problems, making it a highly relevant future tool, as explained below:

1. Already within the magnetic field and temperature operation range reported in the current paper there are important phenomena that can be optimally studied by a scanning qubit. The qubit operation requires a parallel magnetic field ($>3\text{T}$ reported in this paper). If the qubit scans a 2D system, this field can be made parallel to the 2D system, and therefore to couple only to its spin degree of freedom. The associated Zeeman energy

($g^* \mu_B \sim 350 \mu\text{V}$) is quite small, and would not affect a large variety of interesting 2D phenomena.

2. It is possible to independently add a magnetic field perpendicular to the 2D system. We believe that this field will not strongly affect the operation of the qubit, but will allow to study the behavior of the 2D system even in the limit of strong perpendicular fields that are not accessible by e.g. scanning SQUIDs and scanning NV centers.
3. We note that since the qubit lines appear over a large range of magnetic fields, once the field is above the required minimum field the qubit can operate continuously as a function of increasing fields.
4. There is a large variety of interesting phenomena to be studied within the above magnetic fields and temperatures. For example, we have shown that a SET can image the flow of ballistic electrons in graphene (Nat. Nano. 14, 480, 2019). These experiments were done in the semiclassical flow regime, where the quantum coherence of electrons did not play any role. The improved qubit sensitivity, by a factor of 10 over that of SETs, now allows working at much lower source-drain voltages and reach for the first time **quantum** flow phenomena. These include various electron optics phenomena, electron interference, electron localization, magnetic focusing of electrons and of composite fermions, the imaging of valley hall effect due to berry curvatures, and this is only a small fraction of the possibilities within the demonstrated field and temperature range of the qubit. Of course this is not limited to graphene but can be performed on other 2D systems.
5. To give one concrete example, we can mention the recent imaging experiments that we performed on magic angle twisted bilayer graphene (arXiv 1912.06150), which discovered key features of this system. Most of the measurements in that paper were done in a large parallel magnetic field and at low temperatures, fully compatible with the operation of the qubit. If we could have done these experiments with a qubit rather than a SET it would have taken us 1/100 of the measurement time.
6. Another important advantage of the qubit over SET that was not emphasized enough in the paper is its minimal back-action (invasiveness). A measurement with a SET involves single electron current flowing through the sensor, namely, the charge state of the detector has to fluctuate between N and $N+1$ electrons. These charge fluctuations can have a strong backaction, especially when trying to image single electrons or fragile quantum phases. In our recent paper on imaging the Wigner crystal of electrons (Science 364, 870 (2019)) we worked very hard to prove to ourselves that this backaction had a negligible effect on the imaged Wigner crystal. In contrast, measurements with the charge qubit do not involve fluctuations of its charge state. Instead the electron just changes its wavefunction. This leads to significantly smaller backaction. In fact, by controlling the extent of the wavefunctions, we can continuously trade sensitivity for reduction in back-action. In the Wigner crystal experiments, sensitivity was never a problem - it is very easy to 'see' single electrons. The main problem was to avoid the backaction. The fact that backaction in qubit measurement can be significantly reduced is a key feature that makes it extremely useful in imaging a variety of other fragile quantum phenomena.
7. In fact, it should be possible to extend the operational temperature range of the qubit to higher temperatures. This can be achieved by heating only the sample and keeping the qubit cold. This is how the experiments on electron hydrodynamics at elevated

temperatures were performed (Nature, 576, 75, 2019). There we heated the sample to 150K using a resistive heater attached to the sample and decoupled from the SET, and kept the SET at 4K. Similar thing can be done with the qubit. In principle it should be possible to do this sample heating with minimal power by heating just the electronic system with currents (as was beautifully demonstrated recently by Josh Folk in an experiment measuring the entropy of a quantum dot, Nat. Phys. 14 1083 (2018)). With this technique it should be possible to keep the qubit at <100mK while heating the electrons in the measured sample to tens of Kelvins.

8. In the paper we demonstrated the operation of the qubit at a parallel magnetic field larger than 3T. We believe that this minimal field could be significantly reduced, based on our previous experiments in Cornell (see Ferdinand Kuemmeth thesis, Cornell University) with very short (~100nm) quantum dots in clean nanotubes. There we could see the same dark/bright transitions already at fields of >1T, suggesting that lower fields can be used to form the qubit using the mechanism discussed in the paper.

In summary, there are plenty of interesting imaging experiments that could be performed with the new qubit detector. Such experiments would be of broad interest to the van-der-Waals and mesoscopic physics mesoscopics communities at large. We realize that in the original paper we did a poor job in explaining the above message and thank the referee for this question. We amended this in the revised version of the manuscript and Supp. Info.

Changes to the manuscript: In the discussion paragraphs in the paper we 1) added a sentence describing the generally small effect of a parallel field on 2D system and the possibility to work in perpendicular field 2) mentioned that samples can be measured at higher temperatures by thermally decoupling them from the qubit. 3) Described the crucial importance of reduced back-action of the qubit 4) Expanded the list of possible phenomena that the qubit will enable to image

Is the work convincing, and if not, what further evidence would be required to strengthen the conclusions?

The scientific evidence is quite compelling. However, the authors may consider showing the power spectral density of the measurements in far-detuning and non-detuning modes of SET vs Qubit operation. Seeing the spectral density across the measurable frequency range of the sensor would give a sense of the wider applicability of such a sensor.

We thank the referee for this question. Indeed we did not present clearly the frequency dependence of our detected signal. We have added a new section to the supplementary information that describes this in detail. The explanation is also reproduced below.

The figure below shows the spectrum of measured current for $\Delta V_g = 0$ (zero detuning), $\Delta V_g > 0$ (far detuned, red curve) and $\Delta V_g < 0$ (far detuned blue curve), shown in a narrow window around the down-converted frequency $f \approx 1.45$ MHz, for RBW=20 Hz. The noise floor values in all three measurements coincide. The wide peak over the entire window range corresponds to the resonance peak of the LC tank circuit used in the measurement. Ultimately, the frequency band in which this measurement can be performed is limited by the LC tank in our experiment. In

addition to the common noise floor, additional parasitic signals are seen in all three curves, as a result of the measurement set-up and protocol (blue arrow - DC offset noise, black arrows - high harmonics of gate ramping, measured due to capacitive crosstalk). The main difference in the noise spectra between the three detuning points is observed near the frequency at which the excitation on the source contact is introduced (red arrow): very different conductance, seen as a δ -function in the spectrum for the zero-detuned case, and a skirt of elevated noise floor around it resulting from shot noise of the measured conductance over each measurement period. This effect is seen for very long period times (in this case, 50 μ s), and decreases for shorter times, as explained in Supplementary Information S9. Ultimately, for short enough measurement periods, the noise floor is completely insensitive to detuning, and is currently limited by the noise floor of the current preamplifier and not by charge noise, as seen in Fig. S7.

This clearly shows that there is still a place for improving the sensitivity: by decreasing the resistance of the contacts to the nanotube, the transconductance of the nanotube will be improved, and correspondingly the voltage signals will be amplified more. In this way we will obtain better voltage/charge sensitivity, which would be ultimately limited by the charge noise of the system, a limit that has not reached in the current measurements due to the large contact resistance.

Changes to the manuscript: A new section has been added to the revised Supplementary Information (S11) which includes the above explanation and figure.

On a more technical note, if the lever arm for each gate falls in the range of <0.2 , then does that suggest a non-zero cross-capacitance from the 7 gates to separate parts of the CNT? If so, how is that taken into account when mapping out the spatial profile of the Bright and Dark wavefunctions?

No this doesn't mean that. The lever arm gives the ratio of the gate capacitance to the total capacitance and does not reveal the cross capacitance between a gate and its neighboring sections on the nanotube. Since we have 7 gates in this device, by definition, the lever arm of

each must be smaller than $1/7$. In fact, in our device the suspension height of the nanotube above the gates (60nm) is much smaller than the distance between gates (140nm). To zeroth order this means that the gates act mostly locally, and that their cross capacitance to segments of the nanotube above their neighboring gates is quite small. But indeed, there is a small effect that we have measured and quantified carefully in this paper. In fact, this cross coupling effect reduces the spatial resolution of our wavefunction imaging in figures Fig. 2c,d ,Fig.3e, since each pixel now gets slightly ‘smeared’ to the next neighbors. Since we know the cross capacitances rather accurately, we could deconvolve this smearing. In the main text we presented the raw data without attempting to do this deconvolution, and already there, the difference between the charge density of the localized dark and extended bright states could be clearly seen. In the SI of the paper we showed that by taking into account the cross capacitances we can in fact improve the resolution of our imaging and resolve that the localized dark state is significantly narrower than in the raw data by about a factor of two.

Changes to the manuscript: To make this point clearer to the reader we have added a new SI section (SI3) that explains the above.

Also, if the qubit state is as robust as the authors claim, it’s unclear to me why the lifetimes ($T_2^* \sim 0.9\text{ns}$ and $T_1 \sim \mu\text{s}$) are so short. Have the authors quantitatively studied the effect of magnetic instability and high contact resistances on such values? I’m aware that there’s a supplemental section on the theoretical limit on the potential sensitivity, but there’s no mention of what would be the expected T_2^* and T_1 of such an ideal scenario.

Since this is a charge qubit, the most likely bound on T_2^* would be given by charge noise. It should be noted that in many other experiments on charge qubits similar coherence times were obtained. The expected coherence times in this case depend on electric noise in the environment of the qubit, due to surface charges and instrumentation noise. Due to significant contribution of localized sources of noise (on the scale of individual gates), we have no independent way of estimating these noises rather than by the qubit measurement itself.

Current experimental limitations (magnet with no persistent current switch) limit the stability of the magnetic field during the measurements to $\sim 25\mu\text{T(rms)}$, estimated from magnet control feedback loop measurements (shown here for $B=3\text{T}$):

These fluctuations impact the obtained sensitivity of magnetic field measurements, and electric field measurements (through the relation between magnetic and electrical moments of the qubit) and set a limit for maximal dephasing time T_2^* which will translate to $\sim 15\text{ns}$ in our setup. Additional experimental factors, such as fluctuations of the magnetic field on shorter time scales, not registered by the controller, and local charge noise near the device, will limit the result further.

We agree that these are important details that were missing in the original version of the manuscript and have therefore added them to the modified Supplementary Information.

Changes to the manuscript: We added a new SI section (SI10) describing the measurements of the magnetic field fluctuations and to what extent this can limit the observed coherence times.

On a more subjective note, do you feel that the paper will influence thinking in the field? Please feel free to raise any further questions and concerns about the paper.

I believe the technical feat presented in this work is very impressive, and it's quite possible that such a system will be applicable to studies in the future that are not yet clear to the community now. The novelty and technical feat already justifies publication. While the community would benefit from seeing this technological progression of gate-defined quantum dots in CNTs, it's not entirely clear to me that such a demonstration would be of immediate interest to the general readership of Nature Communication.

In the previous answers we explained why the new qubit can serve as a rather generic platform for imaging a variety of interesting physical systems. The introduction of quantum sensing into electrical field imaging, which was so far done via classical devices such as a single electron transistor, is a fundamental step-up for the field on its own. Furthermore, in the answer to the next question below we also explain that our qubit can be implemented using the simplest single-gated nanotube transistor device structure, which is routinely made by more than a dozen of groups. This makes it directly relevant for a broad range of groups working with nanotubes. Specifically, we think that several groups that work on nanomechanics in suspended nanotubes can directly benefit from this new qubit in experiments that aim to reach the quantum limit of mechanics.

Finally, we want to stress that our paper has two important and independent results: the first is that the nanotube qubit can be used as a scanning sensor with improved electrical field detection sensitivity and capability to detect magnetic fields. But a completely independent result, which is an important achievement by itself, is the creation of a completely new type of a charge qubit in a semiconducting systems. Many works have studied charge qubits implemented in double quantum dots. In this paper we created for the first time (to the best of our knowledge) a charge qubit in a single quantum dot, which is based on the natural wavefunctions. Conceptually this new type of qubit is similar to atomic-defect qubits, however, here instead of natural atoms we use a gigantic 'artificial atom' (single quantum dot in a carbon nanotube) with highly extended wavefunctions. This is in a sense very similar to 'Rydberg atoms', which due to their highly extended wavefunctions are so useful for detection and for generating interactions. We believe that just the construction of this new qubit, and the new information it provides on the nature of wavefunctions of nanotubes is important enough for the audience of Nature Communication. For example, no one ever imaged the K and K' wavefunctions and demonstrated directly that in magnetic field they can have dramatically different shapes.

We understand that this more fundamental part of the paper somehow got lost in the discussion about sensing sensitivity, and we believe that it is important to emphasize it as well. To amend this and clearly show that this paper reports two independently important aspects, we decided to

change the title of the paper to 'A new type of charge qubit in a carbon nanotube, enabling electric and magnetic field nanosensing'

We would also be grateful if you could comment on the appropriateness and validity of any statistical analysis, as well the ability of a researcher to reproduce the work, given the level of detail provided.

Perhaps this is discussed in a previous paper by the authors, but it'd be helpful to include some notion of the fabrication yield for producing such CNT-based sensors within a single chip. Or, if there is no way to quantify this yield, how does this challenging fabrication process affect its prospects for wider adoption?

We would like to emphasize an important point that was probably not clear enough in our manuscript: Although the device studied in the paper was assembled using our rather complex nano-assembly technique, a crucial message of this paper is that this **nano-assembly technique is not needed to create the qubit**. In fact, **the qubit will work as well in a simple nanotube transistor devices with a single gate, which are easy to produce and routinely fabricated by more than dozen groups**. The multiple gates in this paper were used to image the basis wavefunction of the qubit and prove the physical mechanism behind its operation, but they are absolutely not needed for making an operational qubit. We think that this makes our proposed qubit directly useful for many groups in the field, for example for those who study nano-electro-mechanics in nanotubes. We understand that this key point was not clear enough in our manuscript and we thus revised our manuscript to explain this better.

Changes to the manuscript: We added new text to the eighth paragraph of the main text that emphasizes that the qubit that we demonstrated in this paper can be implemented in simple single-gated nanotube transistor devices that can be fabricated using standard techniques.

Some typos:

Page 6: "thus, using the same method we can [also] image the spatial..."

Page 6: "this will lead to a shift [in] the Coulomb blockade..."

Page 8: Similarly to Fig 2, we can [directly] image the charge..."

Page 10: "Our sensor requires finite B_{\parallel} for its operation[;] however, it can..."

We thank the referee for pointing out these typos, which we have now corrected.

Reviewer #2:

In this manuscript, the authors propose a scanning probe technique that involves a new type of qubit in a carbon nanotube. This qubit relies on the very special electronic configuration of carbon nanotubes where the valley degeneracy can be split by a parallel magnetic field. This system is thus sensitive to both electric and magnetic fields. The difference of DC conductance in the two qubit states allows a simple readout scheme, relying only on conductance measurements. In this paper, the qubit is characterized and its sensitivity as a sensor is estimated. It proves to be more

sensitive to electric field than SET scanning probes and comparable with NV centers for the magnetic field.

On the whole, the article is clearly written and succeeds to explain the principle of the detection. The data presented are clean and look sound. Additional details on various aspects of the experiment are present in the supplementary information. The conclusion of the article, namely that this new kind of qubit is a valuable tool for quantum sensing, is rather convincing and will probably be interesting for the community.

Note that there is no data demonstrating the use of this qubit for sensing of a “real” physical system, such that the point of the article is more applied than fundamental.

I thus recommend publication in Nature Communications, providing that the authors address the minor following points:

We thank the referee for appreciating the importance of our work and for recommending its publication.

- Very few details are given about the fabrication of the sample. The authors refer to a previous work, but it would be beneficial to give some important information in the paper, at least in the supplementary information. For example, what is the material contacting the carbon nanotube? This would be interesting regarding the contact resistance.

This is an important comment and we have now amended the manuscript to include this information. Independently, we would like to note that although the device that was studied in the current work was assembled using our rather elaborated nano-assembly technique, an important message of this work is that this qubit can be formed equally well in the simplest single-gated nanotube transistor devices that have been fabricated by more than dozen of groups in the world. In this sense, one could use other recipes that have been published in the literature to obtain a similarly good qubit. This message was not stressed clearly enough in our original manuscript, so in addition to adding our fabrication details to the revised supplementary information we also explained this message.

Changes made to the manuscript: We added new text to the eighth paragraph of the main text that gives more details regarding the fabrication of our samples, but at the same time emphasizes that the qubit that we demonstrated in this paper can be implemented in simple single-gated nanotube transistor devices that can be fabricated using standard techniques.

- The setup relies on the fact that the valley is a good quantum number, which is still the case in presence of spin-orbit coupling (page 4). However, it is quite common to find mixing between the K and K' valley, even in very clean nanotubes due to boundary conditions (Marganska, PRB 92 (2015)). How difficult is it to find a “good” carbon nanotube? Does this compromise the relative simplicity of the setup?

That is an excellent question. We have tested many different K-K' crossing vertices in few different cooldowns of the same device. In each cooldown the device behaved somewhat differently in transport, mostly because the workfunction of its contacts was different due to a difference in the vacuum pump-down time. Yet, in all cases we found that about 1/4 of the K-K' vertices allowed creating a good qubit, with performance that in many cases was similar to that described in the paper.

The referee correctly noted that due to boundary conditions the K and K' might not remain good quantum numbers. We believe that the fact that we observe good qubit operation in a large number of K-K' vertices suggests that a) K-K' mixing in our devices is rather weak, and b) the working principle of the qubit survives also weak K-K' mixing.

There are four requirements for the 'atomic-like' qubit to work properly:
1) It should have two basis states that are roughly orthogonal, namely the tunneling element between them should be small.

2) These states should have a large difference in their conductance (dark/bright state). This is crucial for the simple transport readout that we showed in the paper.

3) The states should have significantly different charge distributions to allow electric field detection.

4) The states should have different magnetic moments to allow magnetic field detection.

Our experiments measure these four properties for each vertex independent of model interpretation. The magnetic moment difference between the two basis states (point 4) is seen directly from the slopes of the dark and bright Coulomb peaks in magnetic field. The difference in charge distribution (point 3) is probed directly by the difference in response to local gating. The difference in conductance (point 2) is seen already in DC transport by observing dark and bright states with dramatically different conductance (e.g. fig 1e), and is obtained independently from time domain measurements (e.g. as in fig 4c). The fact that the two states are orthogonal can also be seen directly in DC transport (given by the rounding of the anti-crossing) and of course can be obtained directly from time domain experiments. We indeed find that in many vertices these four properties are nicely obeyed. Specifically, we see that in large fraction of the vertices the two states have very weak mixing, which as we explained in the paper comes from the fact that not only their K and K' wavefunctions are roughly orthogonal but also are their spin orientations. Moreover, we consistently see that K-like states are bright in transport and K'-like states are dark. The difference in the conductance of these states often differs by many orders of magnitude (We note that we have already observed similar systematic dark-bright behavior in measurements that we performed in Cornell (see thesis by Ferdinand Kuemmeth, ref. 29 in our paper)). So experimentally we find that these conditions are obeyed consistently in large fraction of the vertices, which suggests that the effect of K-K' mixing is weak and not detrimental to the qubit operation.

Changes made to the manuscript: This question touches on an important physics point, and we think that the discussion above can indeed give the reader a more complete picture of the

generality of the experimental observations. We therefore added a concise version of the above discussion to the revised Supplementary Info S14 (second part of this section).

Here are a few typos I found:

- Caption of fig. 1.b : a space is missing between “into” and “the”
- Page 5 “The nanotube is suspended a distance of 1.2 μ m”: I think a word is missing
- Caption of fig. 5: DC electric potential measurement

We thank the referee for noting these typos, which are now corrected.

Reviewer #3:

The paper "Electric and magnetic field nano-sensing using a new, atomic-like qubit in a carbon nanotube" reports the implementation of a new quantum sensing scheme using transport measurements in an ultra-clean carbon nanotube device. Quantum sensing is a very active field of quantum technology in which the idea is to use the control on individual quantum system as an ultimate probe -ideally quantum limited- for various applications. In particular, scanning probes using such systems are very interesting for probing a variety of condensed matter states.

The manuscript describes a conceptually new scheme which uses a rather simple system - single dot in a carbon nanotube with an easy electrical interface. This new "atomic-like qubit" is made possible by the nano-assembly technique pioneered by the group of S. Ilani for making carbon nanotube quantum devices. Carbon nanotubes are very appealing in that context since they are a very good match between mesoscopic devices and very well controlled spectrum. The results demonstrated here provide a proof of principle that this new quantum sensor has a high combined sensitivity to local electric and magnetic fields. It could therefore be very interesting to incorporate it in an actual scanning probe setup.

The results presented here are definitely of high quality and the manuscript well written. I recommend publication in Nature Communications.

We thank the referee appreciating the importance of our results and recommending publication in Nature Communications.

I would like the authors to consider the following comments/questions:

1. I understand that the device presented in the paper is a proof of principle and that probably many technical developments will have to be sorted out to incorporate it in a scanning setup. However, it would be nice to have more details about how the authors technically envision this, in particular regarding how to bring the K_n/K_m to degeneracy.

We have now added into the revised manuscript a more detailed description of the technical requirements for integrating the qubit in a scanning probe microscope setup, which we also repeat below:

Currently our scanning SET probe setups works at low (acoustic) frequencies. Integrating the new qubit measurement modality requires few additional changes:

- a. High frequency cabling to the microscope. This is not trivial, but has been achieved in several scanning setups (e.g. NV centers, scanning microwave, etc.). High frequency cables are often more rigid than regular coaxes and therefore can exert larger forces on scanning piezo stages. We note, however, that for the measurements done here (e.g. fig 5) the cables need to connect only to the cantilever and not to the sample. Thus, if the sample is mounted on the piezo motors and the cantilever is fixed, we avoid the above problem altogether.
- b. Vector magnet - the qubit requires magnetic field parallel to the nanotube axis. Ideally, we would like to also have an independent control over the perpendicular field. This field is often the one that most strongly controls the physics of a potential 2D sample under study. Addition of perpendicular field can also improve the performance of the qubit (see answer to question 2 below). This independent control of fields would be best achieved using a vector magnetic field.
- c. Persistent mode magnet - the measurements in the current paper were done in a dilution fridge with a magnet that did not have a persistent switch. The magnet is driven by a power supply fitted with low noise circuitry, which measures the current in the copper leads and stabilizes it using feedback. We found that even with this setup, the remaining magnetic field fluctuations bounded the qubit performance. To eliminate this source of noise it would be beneficial to have a magnet with a persistent switch fitted.

The process of tuning the device is indeed more complicated than tuning the device in SET modality. In addition to locating Coulomb blockade peaks, we need to trace their positions as a function of $B_{||}$, and locate the switching between different wavefunctions, as a switch between visible CB conduction lines for K states and invisible lines with opposite slope for K' states (see Fig. S7). For each switching point in $(V_G, B_{||})$, we check whether a slow tunneling rate between the two switching states is observed, indicating a spin flip. This is done by performing the time domain sequence in Fig. 3a to observe whether sharp lines of decreased conductance are observed (Fig. 3c). While overall the process is time consuming, it needs to be done only once to allow calibrating the device as a sensor, and thus will not be a limiting factor for scanning measurements with the qubit.

Changes to the manuscript: Added a description of the above considerations to SI section S14.

Have the authors also thought about the back-action of this sensor on some benchmark physical systems ?

This is an excellent question, which indeed was not discussed in the paper. In fact, the answer to this question highlights an important advantage of the qubit as a scanning probe that was not emphasized in our manuscript. We have added a paragraph to the revised manuscript that explains this in more detail, and also give the explanation below.

The original version of the paper discussed mostly the sensitivity of the qubit sensor, which is the important figure of merit when probing robust electronic ground states. However, many interesting states of matter are fragile, and can be strongly affected by the backaction of the detector, and then the sensitivity is not the only important factor. A measurement with a SET involves single electron current flowing through the sensor, namely, the charge state of the detector has to fluctuate between N and $N+1$ electrons. These charge fluctuations can have a strong backaction, especially when trying to image single electrons or fragile quantum phases. In our recent paper on imaging the Wigner crystal of electrons (Science 364, 870 (2019)) we worked very hard to prove to ourselves that this backaction had a negligible effect on the imaged Wigner crystal. In contrast, measurements with the charge qubit do not involve fluctuations of its charge state. Instead the electron just changes its wavefunction. This leads to significantly smaller backaction. In fact, by controlling the extent of the wavefunctions, we can continuously trade sensitivity for reduction in back-action. In the Wigner crystal experiments, sensitivity was never a problem - it is very easy to 'see' single electrons. The main problem was to avoid the backaction. The fact that backaction in qubit measurement can be significantly reduced is a key feature that makes it extremely useful in imaging a variety of other fragile quantum phenomena.

Changes to the manuscript: Added to the manuscript a discussion about the difference between the back-action of a SET and that of the qubit, and how the latter can be continuously tuned to trade sensitivity for reduced invasiveness.

2. What is the physical origin of the avoided crossing Δ ? It does not seem to be mentioned anywhere in the paper.

We thank the referee for this question. Indeed we did not elaborate this very clearly in the text and have now added an expanded explanation to Supplementary Information S5. As mentioned in the paper, the two states that we choose for the qubit have opposite spins. We believe that the very small avoided crossings between them originates from the tunneling between these two opposite-spin states, facilitated by small perpendicular component of the field in our experiment, B_{perp} , which originates from slight misalignment of the nanotube axis with respect to the magnet axis. This perpendicular component introduces a $g\mu_B \sigma_x B_{\parallel} \theta$ term in the Hamiltonian, where σ_x is the X Pauli matrix in spin space, and θ is the misalignment. This term leads to slight tunneling between the states $K \uparrow$ and $K' \downarrow$. In fact, we believe that increasing further the tunneling element between the basis states by introducing an even larger B_{\perp} will most likely improve the performance of the qubit beyond what we reported in the paper. It is well known that the T_2^* time of double quantum dot qubits can improve significantly if it is operated at the 'sweet-spot' of detuning smaller than the tunneling. When the tunneling element is larger than the charge noise, working in the 'sweet-spot' makes the qubit only quadratically sensitive to charge noise. For double dots in carbon nanotubes it was shown for example in Penfold-Fitch et al., Phys. Rev. App. 7 054017 (2017), that the T_2^* time improved in that work by almost two orders of magnitude when operating in the sweet spot (from 300ps to 5ns). Our qubit demonstrated a $T_2^* \approx 0.9\text{ns}$ even when it was away from the sweet-spot. Since our tunneling matrix element was very small, in our experiment, it was not possible to benefit from sweet-spot dephasing improvement. Increasing

the tunneling element (by adding a B_{\perp}) could therefore significantly improve our T_2^* , possibly improving our ability to perform coherent manipulations. Our existing experimental setup did not have a vector magnet and therefore did not allow to test this, but it is an interesting question to study in future experiments.

Changes to the manuscript: We added a concise form of the above to the Supplementary Information section S9.

3. In figure S7, there are clearly additional lines which show that the system departs from the simple picture shown in figure 1b. This could mean (for example if it comes from residual disorder) that the corresponding transition will have an electrical dipole and not only higher order multipoles. It might therefore be sensitive to the non-local part of the electric field. Is it going to be a limiting factor for local electric field sensing ?

This is indeed the true. While many K-K' vertices show clean single-line transition, a significant fraction of them show a more complicated structure of multiple lines. We did not investigate these multiple-line transitions in details, as they seem to be less repetitive and likely to result from details of the disorder on the nanotube. Phenomenologically, however, we see that we can also use these vertices for sensing, and find that the detection sensitivity achieved with these multiple-line vertices is not different than the one achieved in 'simple' single-line vertices. So although the physics in these vertices is less controlled and not fully understood, they still seem to work well as qubit sensors.

Changes to the manuscript: note added to Supplementary Information section S12.

4. The high ohmic resistance of the devices made using the nano-assembly technique seems to be a recurrent problem and is a limitation for the sensing capabilities of the atomic-like qubit. How could this be optimized ?

Indeed, as we mentioned in our previous publications, while our nano-assembly technique yields very clean electronic behavior on the suspended part of the nanotubes, its main drawback is that it does not provide ideal contacts. This is the reason why most of our recent experiments (e.g. attraction by repulsion, Wigner crystal imaging, etc.) did not use transport through the tube but rather charge detection to bypass this problem. We did get considerable improvement in the contact resistance by improving the vacuum level in our assembly system from $\sim 1e-6$ to $\sim 1e-7$, but our devices are still limited to resistances of the order of 100kOhm, which is still far from the ideal quantum resistance. This finite contact resistance currently limits the sensitivity of our scanning SETs and as was explained in the Supplementary Information of the current paper also limits the sensitivity of the qubit measurements. Although we don't have yet a clear solution for this problem, we have been experimenting in the last year with methods to improve the contact resistances by local heating of the contacts, to few hundreds of degrees. This is still very preliminary, and we have very little data-points on that, but so far it looks as a very promising route for significantly improving the contact resistances toward the quantum resistance.

It is important to stress though, that our nano-assembly technique is not necessary for forming the atomic-like qubit, and it should work equally well even with a simplest single-gated nanotube transistor devices that are made by other, more conventional fabrication techniques.

Changes to the manuscript: We added a passage to device description paragraph on pg. 5 to emphasize that our proposed qubit can be made using simple fabrication techniques, and thus should be easily adaptable by other groups working in the field.

5. Regarding the B-field sensitivity, how would single spin $1/2$ detection compare to the actual figures of merit of the atomic-like qubit. Ultra-narrow linewidths of single spin in carbon nanotubes limited by nuclear spins have been recently demonstrated in double dot carbon nanotubes and might be useful as well (see T. Cubaynes et al. NPJQI 5, 47 (2019)).

The recent work by Cubaynes et al. is indeed very impressive. We thank the referee for pointing this out, and we have now referenced this work in the revised manuscript. While the spin decoherence rate demonstrated in that work is indeed extraordinary, the sensitivity of using this qubit as a magnetic sensor is rather limited, somewhere in the few mT range (a precise numerical estimate is hard to obtain from the displayed results as there seems to be no mention of the integration time used in Fig. 3d in that paper). The authors explain that this weak sensitivity to external magnetic fields results from the exchange coupling between the electrons and the ferromagnetic leads. In addition, the method proposed in our paper benefits from two extra factors: Having a significantly higher magnetic moment difference between the two qubit basis states (our qubit uses the orbital moment which is 30 times larger than the spin moment in the above mentioned work), and also a high signal to noise ratio of the measurement resulting from the transport blockade mechanism. Overall these factors lead to better magnetic field sensitivity of our atomic-like qubit.

Changes to manuscript: We added a reference to the work by Cubaynes et. al. together with an explanation of the comparison to our work.

REVIEWERS' COMMENTS:

Reviewer #1 (Remarks to the Author):

Khivrich and Ilani have addressed all of my concerns. I'm looking forward to seeing how the community receives their hard work!

Reviewer #2 (Remarks to the Author):

I am convinced by the author's answers and modifications of the manuscript. I thus maintain my recommendation to publish in Nature Communication.

Reviewer #3 (Remarks to the Author):

The authors have provided extensive and very satisfactory answers to the reviewers reports. I definitely recommend publication of the work in Nature Communications.